# Deep optoacoustic localization microangiography of ischemic stroke in mice

Xosé Luís Deán-Ben [1,2] ✉, Justine Robin [1,2], Daniil Nozdriukhin[1,2], Ruiqing Ni[1,2,3], Jim Zhao[1,2], Chaim Glück [4], Jeanne Droux[3,5], Juan Sendón-Lago [6], Zhenyue Chen[1,2], Quanyu Zhou[1,2], Bruno Weber [4], Susanne Wegener[3,5], Anxo Vidal [7], Michael Arand [1], Mohamad El Amki[3,5] & Daniel Razansky [1,2,3] ✉

Super-resolution optoacoustic imaging of microvascular structures deep in mammalian tissues has so far been impeded by strong absorption from densely-packed red blood cells. Here we devised 5 μm biocompatible dichloromethane-based microdroplets exhibiting several orders of magnitude higher optical absorption than red blood cells at near-infrared wavelengths, thus enabling single-particle detection in vivo. We demonstrate non-invasive three-dimensional microangiography of the mouse brain beyond the acoustic diffraction limit (<20 μm resolution). Blood flow velocity quantification in microvascular networks and light fluence mapping was also accomplished. In mice affected by acute ischemic stroke, the multi-parametric multi-scale observations enabled by super-resolution and spectroscopic optoacoustic imaging revealed significant differences in microvascular density, flow and oxygen saturation in ipsi- and contra-lateral brain hemispheres. Given the sensitivity of optoacoustics to functional, metabolic and molecular events in living tissues, the new approach paves the way for non-invasive microscopic observations with unrivaled resolution, contrast and speed.

New methods enabling breaking through established resolution barriers have massively impacted life sciences ever since first optical microscopes have augmented the visualization capacity of the naked eye[1]. Advanced optical microscopy techniques can now reach extended depths within penetration barriers imposed by the strong photon diffusion in tissues - typically <1 mm within living mammalian tissues[2,3]. Higher resolution has been achieved with electron microscopy[4], while a myriad of technologies based on x-rays, ultrasound (US) or nuclear magnetic resonance have been developed for deep tissue imaging[5]. Yet, the unique ability of optical methods to sense specific molecules in vivo often makes them the preferred choice for biological observations. Recent efforts have thus been directed toward overcoming light diffraction barriers in fluorescence microscopy[6] and developing new optical imaging approaches operating at mesoscopic and macroscopic depths[7,8].

Optoacoustic (OA, photoacoustic) imaging has emerged as a hybrid modality synergistically combining rich optical contrast with superb resolution to provide otherwise unattainable functional and molecular information from deep tissues[9,10]. A number of advanced OA tomographic embodiments enabled breaking through penetration barriers of optical microscopy by providing scalable spatial resolution in the 20–200 μm range at millimeter to centimeter-scale

[1]Institute of Pharmacology and Toxicology and Institute for Biomedical Engineering, Faculty of Medicine, University of Zurich, Zurich, Switzerland. [2]Institute for Biomedical Engineering, Department of Information Technology and Electrical Engineering, ETH Zurich, Zurich, Switzerland. [3]Zurich Neuroscience Center, Zurich, Switzerland. [4]Experimental Imaging and Neuroenergetics, Institute of Pharmacology and Toxicology, University of Zurich, and Zurich Neuroscience Center, Zurich, Switzerland. [5]Department of Neurology, University Hospital and University of Zurich and University of Zurich, Zurich, Switzerland. [6]Experimental Biomedicine Centre (CEBEGA), University of Santiago de Compostela, Santiago de Compostela, Spain. [7]Center for Research in Molecular Medicine and Chronic Diseases (CiMUS), University of Santiago de Compostela, Santiago de Compostela, Spain. ✉e-mail: xl.deanben@pharma.uzh.ch; daniel.razansky@uzh.ch

depths (depth-to-resolution ratio of ~200)[11,12], while further achieving unprecedented 3D imaging speeds in the kilohertz range[13]. OA microvascular imaging is yet restricted to shallow regions, e.g., superficial cortical layers[14], while individual capillaries can only be visualized up to a depth of ~1 mm, where light can be efficiently focused[15,16]. Much like for pulse-echo US, the resolution of OA imaging in deep tissues is chiefly affected by frequency-dependent acoustic attenuation, which effectively reduces the bandwidth of the collected signals, essentially establishing a depth-dependent acoustic diffraction limit. Recently, ultrasound localization microscopy (ULM) broke through this barrier via localization of single circulating microbubbles[17,18]. The basic feasibility of localization optoacoustic tomography (LOT) has likewise been demonstrated by visualizing the flow of absorbing microspheres in scattering phantoms[19,20]. LOT was shown to significantly improve the spatial resolving capacity of OA, while further enhancing the visibility of structures under limited-view tomographic conditions[20], a major deficiency of most OA systems. LOT can additionally provide oxygen saturation readings and dynamic information on the blood flow, namely blood velocity, currently unattainable with conventional OA. Since efficient localization of individual absorbers implies their sparse distribution in the circulation, in vivo LOT is hampered by the strong background absorption of blood. Red blood cells (RBCs) consist of ~$270 \cdot 10^6$ hemoglobin molecules densely distributed in blood (~50% v/v), while typical OA contrast agents, e.g. based on small molecules or nanoparticles, exhibit significantly lower absorbance on a per-unit basis[21]. While individual RBCs can be visualized and tracked by means of optical-resolution OA microscopy using tightly focused light beams[21], such

implementation is limited to superficial tissues where the light can be efficiently focused. On the other hand, individual liquid droplets and circulating tumor cells could be localized in the mouse brain after intracardiac injection using OA tomographic imaging[22,23], but such large (~30 μm diameter) bodies are likely to be arrested in the capillary network, thus hindering in vivo compatibility. Also important is the fact that vascular structures arbitrarily oriented in all three dimensions at different depths cannot be efficiently captured with cross-sectional (two-dimensional, 2D) OA imaging systems, thus three-dimensional (3D) tomographic imaging turns essential for an optimal LOT performance.

Herein, we devised extremely absorbing dichloromethane (DCM) microdroplets with sizes in the order of RBCs. We show that their small size and composition prevents capillary blockage and does not lead to toxic effects, while the strong optical absorption facilitates individual detection and accurate localization within the field of view (FOV) of a volumetric OA imaging system with single shot excitation.

## Results
### Real-time visualization of individual particles in vivo
The basic principle of LOT along with a lay-out of the tomographic approach used to image the mouse brain is depicted in Fig. 1a. A short-pulsed (nanosecond duration) laser tuned to an optical wavelength of 780 nm was used to simultaneously illuminate the entire brain and generate pressure (US) waves via thermal expansion that are eventually detected with a spherical array of transducers (see methods for a detailed description). These pressure waves are produced by endogenous absorbers, mainly hemoglobin in RBC, as well as by the

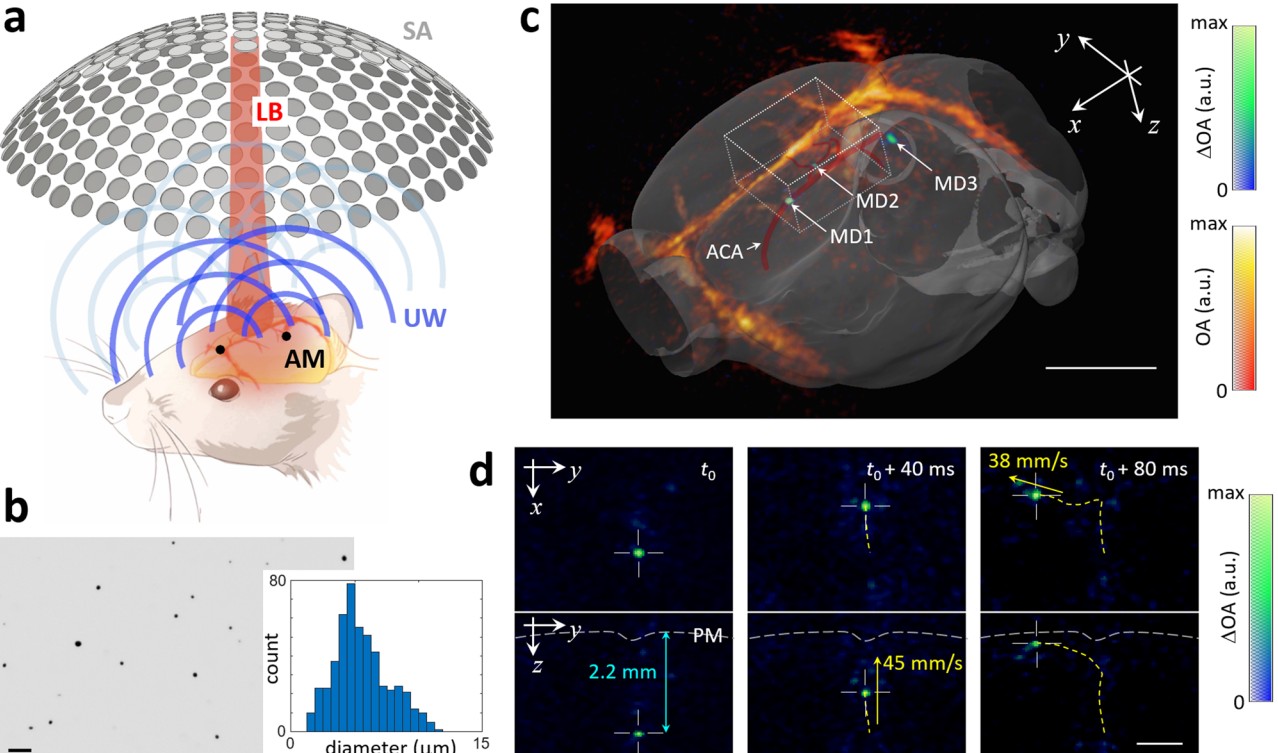

**Fig. 1 | Detection and tracking of individual microdroplets with localization optoacoustic tomography (LOT). a** Lay-out of the experimental setup. AM absorbing microdroplets, LB laser beam, UW ultrasound waves, SA spherical array. **b** Bright field (5×) optical microscopy image of the dichloromethane (DCM) microdroplets along with a histogram of the measured diameters from 500 microdroplets. Scalebar−50 μm. The measurements were repeated 3 times to ensure reproducibility. **c** OA images of the mouse brain superimposed to the differential (background subtracted) OA (ΔOA) images. Three microdroplets in the

azygos pericallosal artery (MD1), middle internal frontal artery penetrating branches (MD2) and pial surface (MD3) are labeled in the differential image. The co-registered Allen mouse brain atlas is also shown. ACA anterior cerebral artery. Scalebar−3 mm, a.u. arbitrary units. Cartesian axes are indicated with arrows. **d** Maximum intensity projections (MIPs) of the differential OA image along the z and x directions for the rectangular box indicated in **c**. Three representative time points are shown. The depth from the pia matter (PM) and the calculated velocities are indicated in blue and yellow, respectively. Scalebar−1 mm, a.u. arbitrary units.

exogenously-administered absorbing microdroplets required for LOT. This leads to a static background OA signal corresponding to vascular structures of different sizes superimposed onto a dynamically-changing signal generated by the particles flowing in blood. LOT can only be performed if the signal generated by an individual (isolated) particle is sufficiently high to be detected in a single image voxel. The background signal per voxel is proportional to the number of RBCs enclosed, which can reach 100 s to 1000 s for the given ~100 µm resolution of the imaging system. The DCM droplets employed have an approximate concentration of 200 mM of IR-780 dye (extinction coefficient ~250×10³ M⁻¹cm⁻¹). As a reference, the extinction coefficient of hemoglobin at 780 nm and concentration in RBC are ~10³ M⁻¹cm⁻¹ and ~5 mM, i.e., the total light absorption in the DCM droplet exceeds that of a RBC by three to four orders of magnitude (Suppl. Note 1). Note that the very high dye concentration results in strong optical attenuation within the droplet that needs to be considered in order to accurately estimate the absorbed optical energy (Suppl. Note 1 and Suppl. Fig. 1). The diameter of the droplets (Fig. 1b, mean: 5.5 µm, standard deviation: 2.1 µm) is smaller than the disk diameter of murine RBCs (~4–7 µm)[24]. Individual droplets could clearly be detected in the OA images after clutter removal with singular value decomposition (SVD) filtering (see "Methods" for a detailed description). Manual co-registration of OA images with the Allen mouse brain atlas[25] (Fig. 1c) further served as anatomical reference to identify the droplets' position at a given time point. For example, three microdroplets in the anterior cerebral artery system (MD1, azygos pericallosal artery and MD2, middle internal frontal artery penetrating branches) and in the pial surface (MD3) are clearly visible in Fig. 1c. The flow of microdroplets is better visualized in two movies available in the online version of the journal (Suppl. Movies 1 and 2). Note that the OA system's ability to cover an entire 3D region with single-shot excitation is of particular importance for an accurate trajectory estimation, e.g. when a droplet follows the blood as it flows from a deep-seated artery into the superficial cortical vessels (Fig. 1d). Note that the high temporal resolution (100 3D images per second) is paramount for accurately quantifying the relatively high blood flow velocity.

### Breaking the acoustic diffraction barrier with LOT

LOT images were subsequently rendered by aggregating the localized positions of the microdroplets flowing through the vascular network over a temporal sequence of OA images (see methods for details). Importantly, the effective integration time of the images used for particle localization must be minimized. In our case, no average or compounding of multiple frames was required for localizing individual particles, which is crucial considering the high blood flow velocity resulting from ca. 200 ml/100 g/min perfusion in the mouse brain, depending on anesthesia and depth from the surface[26,27]. Note that it only takes a single nanosecond laser pulse to generate the entire volumetric OA dataset (single shot excitation). Hence, the captured structures are very well defined in time by the laser pulse duration, facilitating accurate localization of particles moving with high velocity. The image formation process in LOT can be better visualized in a movie available in the online version of the journal (Suppl. Movie 3). The enhanced resolution achieved with LOT (formed with ~6000 tracks) can be seen in the dorsal view of the mouse brain vasculature (Fig. 2a, b) as well as in the rotating 3D view provided in the online version of the journal (Suppl. Movie 4). Pial vessels such as the anterior cerebral artery (ACA), the middle cerebral artery (MCA) and the posterior cerebral artery (PCA), as well as the superior sagittal sinus (SSS) are clearly identified in the LOT image. The penetrating and ascending vessels (long and short cortical branches) propagate across the cortex over ~1 mm (approx. Bregma −0.22 mm to −1.22 mm, Fig. 2b coronal view), as expected[14,28]. Such features cannot be resolved in standard OA images not only because of its inferior resolution but chiefly due to limited-view effects hindering reconstruction of vertically-oriented

structures[20]. Vascular structures separated by 22 µm could be resolved by LOT over a depth range of ~1.5 mm from the brain cortex (~2.5 mm from the skin surface) through the intact ~0.5 mm-thick scalp and 0.4–0.6 mm-thick skull[29,30]. A comparison of the LOT images rendered as a function of the number of localized points and tracks accumulated over time is provided as supplementary information (Suppl. Fig. 2). It is shown that most vascular structures can be resolved by analyzing approximately 500 consecutive frames, i.e., within 5 s acquisition time.

### Blood flow velocity and light fluence mapping

The high temporal resolution of the 3D OA imaging system further enabled estimating the blood flow velocity via particle tracking (see methods section for a detailed description). The reconstructed velocity map (Fig. 2d) is consistent with previously reported values for the mouse brain, i.e. from several cm/s in major cerebral arteries (corresponding to a typical blood perfusion range of 0.5-1.6 ml/g/min depending on the brain region and depth[31,32]) down to <5 mm/s in smaller vessels[33,34]. A rotating 3D view of the velocity map is provided in the online version of the journal (Suppl. Movie 5). It was further possible to accurately assess the velocity profile across relatively large vessels, revealing a Poiseuille-like flow profile (inset in Fig. 2d). Accurate quantification of blood flow represents an important new capacity in the functional imaging portfolio of OA, which is already capable of measuring multiple hemodynamic parameters such as blood volume and oxygen saturation[35,36]. It is also important to consider that the OA signal generated by the microdroplets is directly proportional to the local light fluence provided the linear OA response is preserved. We observed a clear signal decay with depth (Fig. 2e, top, approx. Bregma −1.46 mm). This phenomenon is also evinced by the sudden disappearance and reappearance of droplets, arguably ascribed to their migration to/from the regions exposed to low light fluence. The light fluence distribution could then be approximated by fitting an exponentially decaying function to the measured signal intensities at different positions in this section (Fig. 2e bottom, see "Methods" for a detailed description). Note that the light fluence can still be estimated with polydisperse (non-uniform in size) particles provided that a sufficient number of OA responses from such particles has been captured (Suppl. Note 2 and Suppl. Fig. 3). As tissue composition varies significantly across different brain regions, a heterogeneous fluence distribution is generally expected. To accurately assess which brain regions are efficiently illuminated (Fig. 2e, bottom), we co-registered the LOT images with the Allen brain atlas by identifying vessels in the coronal section, e.g. the pial artery, transverse hippocampal artery/vein and middle cerebral artery. Note also that the light fluence distribution is wavelength-dependent. Thereby, for more accurate extraction of oxygen saturation values the light fluence must be estimated over a relatively broad spectral range, which is not possible with the currently available droplets.

### Multi-parametric super-resolution characterization of stroke

While ultrasound-based localization approaches may similarly render high-resolution microvascular maps of the rodent brain beyond the diffraction limit[18], OA has the complementary advantage of providing functional and molecular information not readily available with other imaging modalities, such as blood oxygen saturation values[12]. We exploited the added value of LOT in a murine model of stroke ($n = 5$). Briefly, acute ischemic stroke was induced in BALB/c mice by occluding the left MCA via thrombin injection to induce a clot formation in situ (see "Methods" for a detailed description). The volumetric OA imaging system was used to collect data both from the left (ipsi-lateral) and right (contra-lateral) brain hemispheres. Two sequences of images following injection of 100 µl of microdroplet emulsion were acquired from each mouse delayed by approximately 30 min. The LOT images revealed a clear reduction of microvascular density in the ipsi-lateral side (Fig. 3a and Suppl. Fig. 4). This was also observed in LOT images of

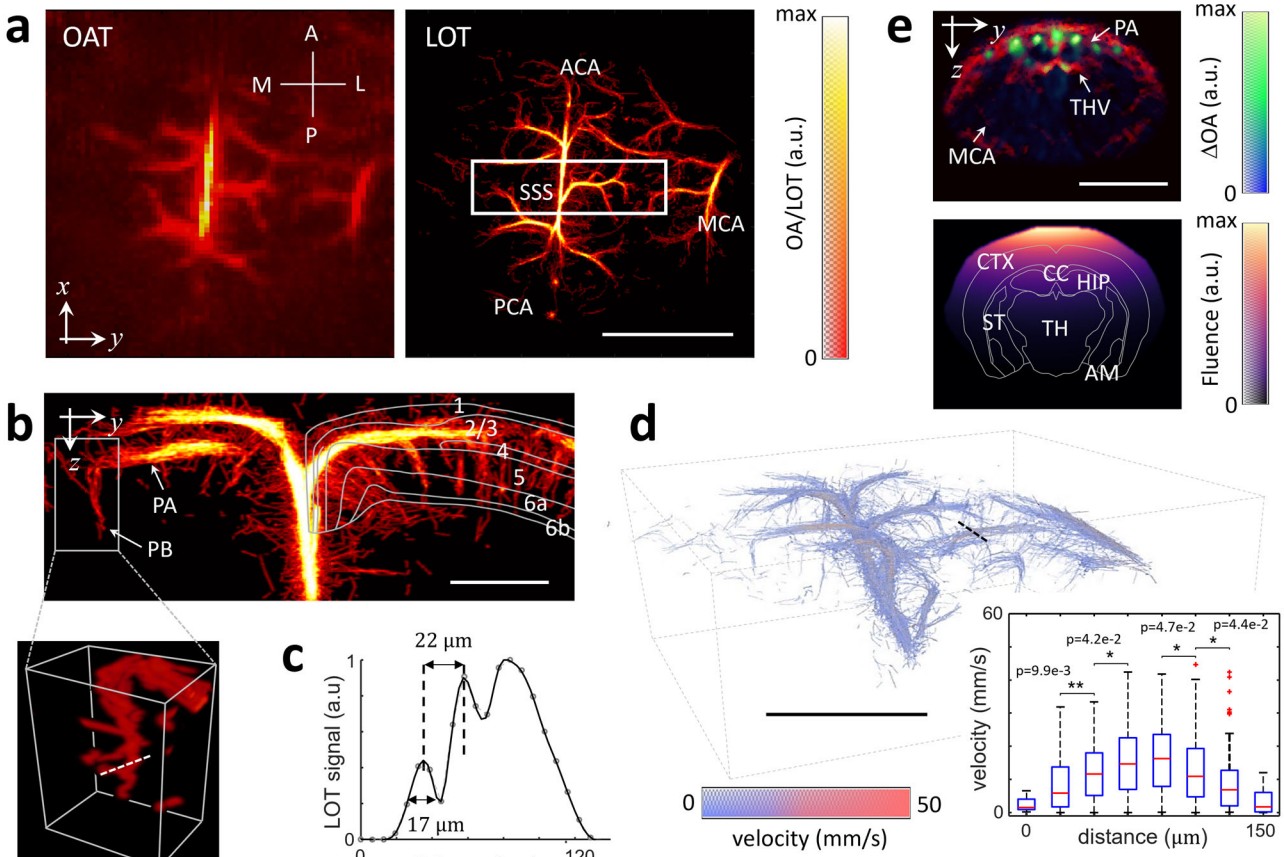

**Fig. 2 | Structural and functional imaging with LOT. a** Maximum intensity projections (MIPs) along the z direction (dorsal view) of the 3D images rendered with conventional OA tomography (OA, left) versus localization optoacoustic tomography (LOT, right). A anterior, P posterior, M medial, L lateral, ACA anterior cerebral artery, MCA middle cerebral artery, PCA posterior cerebral artery, SSS superior sagittal sinus. Scalebar−3 mm, a.u. arbitrary units. **b** MIP images along the x direction (coronal view, Bregma −0.22 mm to −1.22 mm) of the rectangular box of the LOT image indicated in (a). Cortical layers are indicated. The inset displays a 3D view of the indicated region. PA pial arteriole, PB penetrating branches. Scalebar− 1 mm. **c** Profile of the LOT image for the dashed white line in **b**, a.u. arbitrary units. **d** 3D view of the blood flow image. Inset shows boxplots of the measured velocities (40 values per boxplot) along the dashed black line. Outliers are plotted as red squares. Mean+1.26$\sigma$, upper quartile, median, lower quartile, and mean-1.26$\sigma$ are shown. Values for individual droplets are provided in a supplementary file. Statistical significance is shown (*$p < 0.05$, **$p < 0.01$ for one-tailed unpaired $t$ test). Scalebar−3 mm. **e** Cross-sectional coronal view (top, approx. Bregma −1.46 mm) of the OA image along with the superposition of 8 differential OA images (ΔOA). The light fluence distribution image (bottom) is derived by fitting the signals of 50 microdroplets in the differential OA images to an exponentially decaying function. MCA middle cerebral artery, PA pial artery, THV transverse hippocampal vein, CTX cortex, CC corpus callosum, HIP hippocampus, TH thalamus, ST striatum, AM amygdala. Scalebar−5 mm, a.u. arbitrary units.

the same region taken before and after stroke (Suppl. Fig. 5). Note that the three representative examples shown in Fig. 3a demonstrate the good sensitivity of LOT in assessing the stroke severity. While barely any functional vascular structures are visible in the ipsi-lateral hemisphere in the first case, a localized damage is clearly observed in the third case. An overall reduction of vascular density is also observed in the second case, although major vessels appear to be functional. The relative number of localized droplets in selected volumes of interest (VOIs) of ipsi- and contra-lateral sides (orange and green regions in Fig. 3a) was considered as a metric of microvascular density. Note that low numbers of localized droplets may hamper light fluence estimations. The percentage of localized droplets was significantly lower in the ipsi-lateral side (mean: 27.27%, standard deviation: 11.90%) as compared to the contra-lateral side (mean: 72.73%, standard deviation: 11.90%). Reduction of penetrating vessels in the ipsi-lateral mouse cortex was also observed in the LOT images. This was quantified as the relative L1-norm of the Sobel-filtered LOT image for the selected VOIs (Suppl. Fig. 6). The relative number of penetrating vessels in ipsi- (mean: 29.72%, standard deviation: 12.99%) and contra-lateral sides (mean 70.28%, standard deviation 12.99%) appear to be approximately the same as the relative number of localized droplets. The velocity

maps further revealed a reduction of blood flow velocity in the area affected by stroke (Fig. 3b and Suppl. Fig. 7), also observed in LOT images of the same region taken before and after stroke (Suppl. Fig. 5). Specifically, the measured velocity in the visible vessels was significantly lower in the ispi-lateral VOI (mean: 16.05 mm/s, standard deviation 1.31 mm/s) than that in the contra-lateral VOI (mean: 19.97 mm/s, standard deviation: 1.58 mm/s). Finally, clear changes in the bio-distribution of oxygenated and deoxygenated hemoglobin, as revealed by spectral unmixing of multi-spectral OA data, were detected in the area affected by stroke (Fig. 3c and Suppl. Fig. 8). The LOT image was subsequently used as a mask to calculate sO$_2$ values in the visible microvasculature (Fig. 3c). A rotating 3D view of the sO$_2$ map for the third case shown in Fig. 3c is provided in the online version of the journal (Suppl. Movie 6). The measured sO$_2$ values were significantly lower in the ipsi-lateral VOI (mean: 0.3087, standard deviation: 0.0594) than those in the contra-lateral VOI (mean: 0.5453, standard deviation: 0.0293). Note that sO$_2$ values in the contra-lateral side are in line with previously reported observations for healthy microvasculature in the murine brain[37]. Cross-talk associated to oxygenation differences in neighboring microvessels generally results in errors in oxygen saturation readings, particularly for separation distances lower than

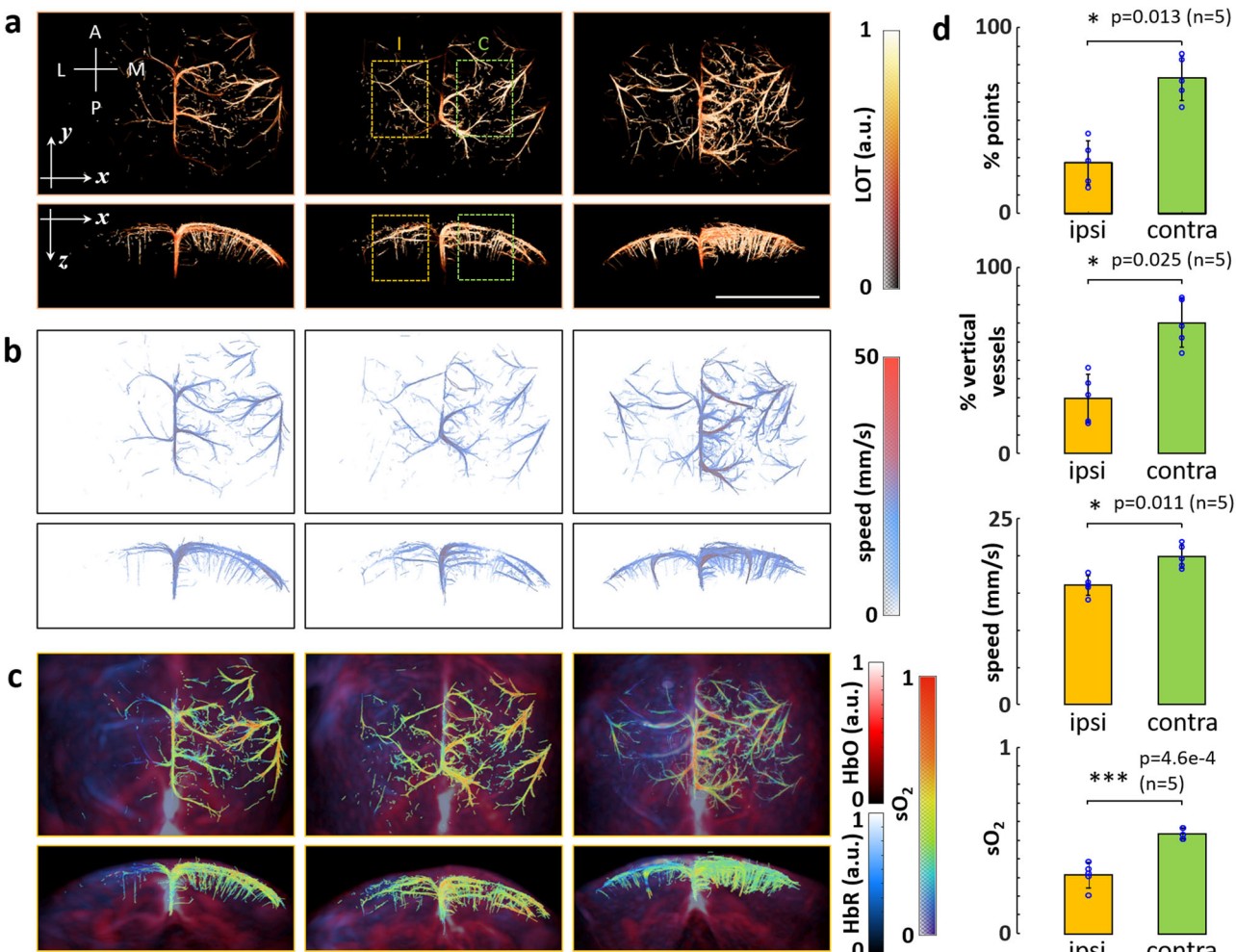

**Fig. 3 | Multi-parametric characterization of ischemic stroke in mice with LOT.**
**a** Maximum intensity projections (MIPs) along the $z$ (dorsal view) and $y$ (coronal view) directions of the LOT images acquired in vivo from three mice affected by ischemic stroke. A anterior, P posterior, M medial, L lateral, I ipsi-lateral, C contra-lateral. Scalebar−4 mm, a.u. arbitrary units. **b** Dorsal and coronal MIPs of the corresponding velocity maps. **c** Dorsal and coronal MIPs of the corresponding bio-distributions of oxygenated (red) and deoxygenated (blue) hemoglobin, a.u. arbitrary units. Oxygen saturation ($sO_2$) values in the micro-vascular structures resolved with LOT are superimposed. **d** Statistical analysis of the measured parameters in selected volumes of interest (VOIs, orange and green regions indicated in **a**) in ipsi-lateral and contra-lateral sides of the brain. The number of mice ($n$) considered are indicated. Individual measurements are plotted as blue circles. First row−Relative number of localized points in ipsi- (27.27% +/− 11.90%) and contra-lateral (72.73% +/− 11.90%) sides. Second row−Relative number of vertical vessels in ipsi- (29.72% +/− 12.99%) and contra-lateral (70.28% +/− 12.99%) sides estimated as the relative L1-norm of the Sobel-filtered LOT image for the selected VOIs. Third row−Blood flow velocities in the visible vessels in the ipsi- (16.05 mm/s +/− 1.31 mm/s) and contra-lateral sides (19.97 mm/s +/− 1.58 mm/s). Fourth row−Oxygen saturation in the visible vessels in the ipsi- (0.3087 +/− 0.0594) and contra-lateral (0.5453 +/− 0.0293) sides. Data are presented as mean values +/− standard deviation. Statistical significance is shown (*0.01 < $p$ < 0.05, ***$p$ < 0.001 for a two-tailed paired-sample $t$ test).

the diffraction resolution limit. However, numerical simulations reveal that it is still possible to differentiate oxygen saturation values in vascular structures beyond the acoustic diffraction barrier (Suppl. Fig. 9). Overall, the multi-parametric multi-scale observations enabled by localization-based and spectroscopic OA imaging revealed statistically significant differences in microvascular density, blood flow velocity and oxygen saturation between ipsi- and contra-lateral sides in ischemic murine brains (Fig. 3d).

**Capillary flow and biosafety of the droplets**
Biosafety of the microdroplets is one important aspect to examine when considering the use of LOT in longitudinal studies and its eventual clinical translation, particularly in regard with capillary occlusion and other potential toxic effects. In the performed experiments, a continuous flow of particles was observed over the entire image sequence without any temporary stall in arterioles and venules,

indicating that no major blockage is induced. This was further verified with two-photon microscopy images of cortical microvascular networks labeled with Texas Red dextran (~400 μm depth, Fig. 4a-b, see methods for a detailed description). Microdroplets appeared with distorted shapes in the DCM-IR780 channel, which was not observed in the images taken before injection (Fig. 4a). This distortion is associated to motion, thus corroborating the undisturbed flow of microdroplets in capillary networks. In rare occasions, circular dots in consecutive slices were observed, arguably corresponding to arrested droplets (Fig. 4b). The number of arrested droplets with respect to the total number of droplets observed in the images was below 1% (Fig. 4c). Similar values were observed when quantifying the number of blocked capillaries (Fig. 4c). Overall, the incidence of these rare occlusion events is comparable to those produced by red blood cells and leukocytes[38,39]. On the other hand, laser speckle contrast imaging (LSCI) of the entire brain following injection of DCM microdroplets not

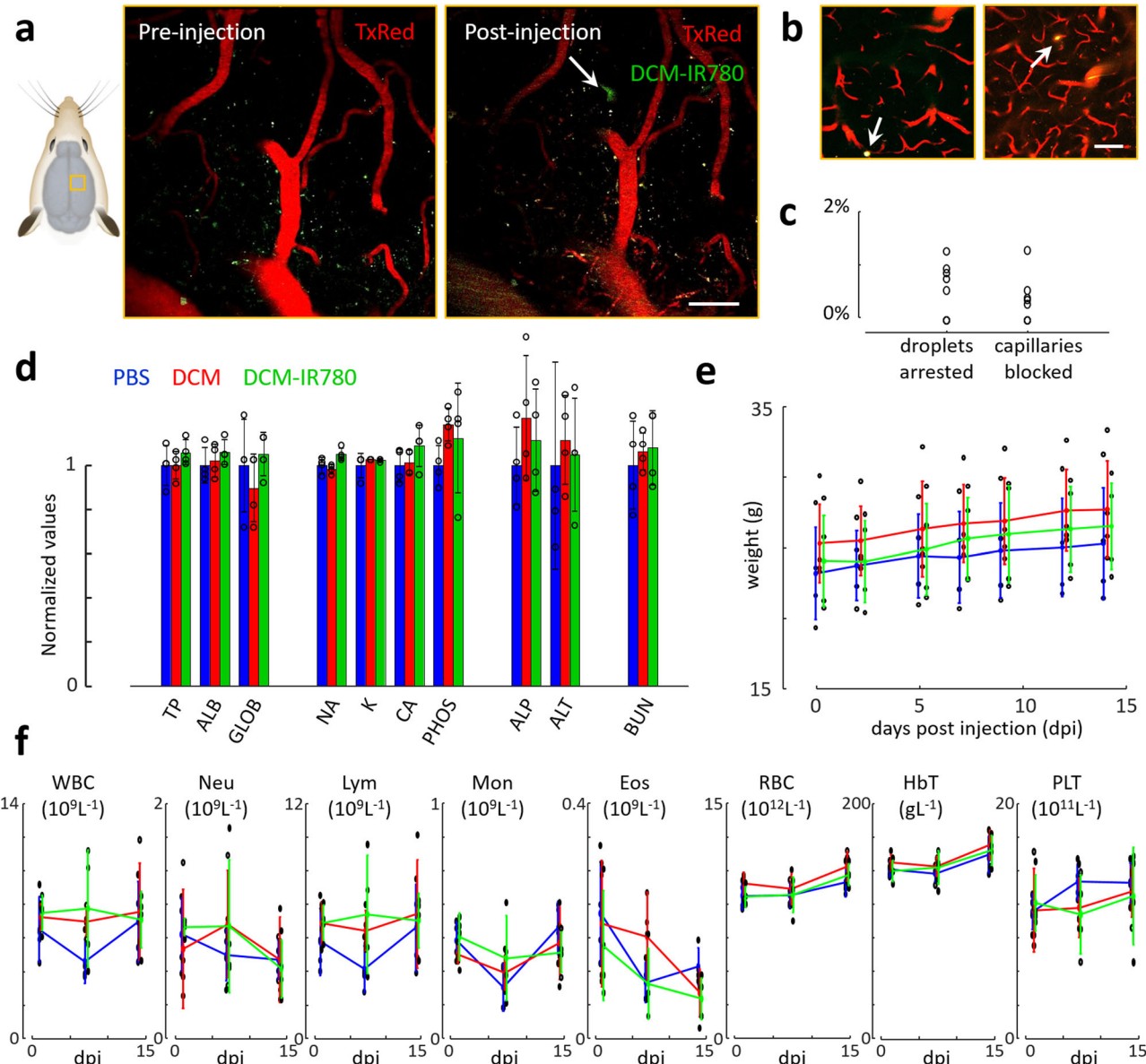

**Fig. 4 | Capillary occlusion and biosafety of microdroplets. a** Two-photon microscopy images taken before (left) and after (right) injection of a 100 µl of microdroplet emulsion. The imaged brain region is indicated. TxRed (607/70 nm filter) and DCM-IR780 (520/50 nm filter) channels are shown in red and green, respectively. The white arrow indicates a microdroplet flowing in the capillary network. Scalebar−50 µm. **b** Equivalent two-photon images for two regions where microdroplet arrest was observed, indicated with white arrows. Scalebar−50 µm. The experiments were repeated for $n = 3$ mice to ensure reproducibility. **c** Relative amounts of microdroplets being arrested in the microvascular network (left) and capillaries being blocked (right) for 8 measurements (black circles) in $n = 3$ mice. **d** Blood biochemistry at day 14 following injection of 100 µl of PBS (PBS, blue, n = 4 mice), 100 µl of DCM microdroplets (DCM, red, $n = 4$ mice), and 100 µl of DCM microdroplets containing IR-780 dye (G3, DCM-IR780, green, $n = 4$ mice).

Individual values (dots) are normalized with the average values for the PBS group, namely 5.45 g/dl total protein, 4.48 g/dl albumin, 0.98 g/dl globulin, 154.75 mmol/dl sodium, 8.28 mmol/dl potassium, 12.53 mg/dl calcium, 14.03 mg/dl phosphate, 107.25 U/l alkaline phosphatase, 30.25 U/l alanine transferase, 15.50 mg/dl blood urea nitrogen. Values for all mice are provided in a Supplementary File. **e** Weight of the mice for the three groups for different days post injection. Values for all mice are provided in a supplementary file. Day 0 corresponds to the pre-injection time point. Dots correspond to individual measurements. **f** Blood hematology at three time points (1, 7, and 14 days) for the three groups of mice. WBC white blood cell count, Neu neutrophil count, Lym lymphocyte count, Mon monocyte count, Eos eosinophil count, RBC red blood cell count, HbT hemoglobin, PLT platelet count. Values for all mice are provided in a Supplementary File. Dots represent individual measurements.

containing IR-780 revealed that no significant changes in blood flow are induced (Suppl. Fig. 10). More quantitative measurements of blood flow velocity performed with two-photon microscopy and with a high speed camera further confirm that blood flow is not altered (Suppl. Fig. 11). The acute toxicity of the microdroplets was also assessed by monitoring three groups of mice over two weeks following injection of 1) DCM-IR780 microdroplets, 2) DCM microdroplets (without dye) and 3) PBS (Fig. 4d–f). No significant differences were observed in biochemical parameters of the three groups measured at the final point

(14 days post injection, Fig. 4d). Total plasma bilirubin and alanine aminotransferase, parameters indicative for functional hepatic impairment and liver cytotoxicity[40], remained unchanged by the treatment. Likewise, plasma urea content as indicator for kidney injury[41] was unaffected. Mice were scored and weighted for a 14-days period following injection (Fig. 4e), and no physical of behavioral changes were observed. Hemodynamic parameters measured at 1, 7, and 14 days post injection were also similar for the different groups (Fig. 4f). Histological sections of major organs embedded in paraffin

wax taken after euthanizing the animals further verified that no tissue damage was produced (Suppl. Fig. 12).

## Discussion

The ability to localize and track intravenously injected micron-size particles in 3D and in the presence of highly absorbing blood background empowers LOT with an unprecedented capacity for microscopic in vivo imaging of deep optically opaque tissues. Breaking through the acoustic diffraction barrier facilitates volumetric visualization of microvascular structures not captured by conventional OA imaging approaches. It was shown that LOT can achieve 20 μm resolution across >3 mm depth range through an intact scalp and skull. Apart from its superb spatial resolution and better visibility of structures under limited-view tomographic conditions, LOT enables quantification of the local light fluence and blood flow velocity. Key to these enabling features are the high 3D imaging frame rate (100 Hz), short duration of the excitation laser pulse (<10 ns) and sufficient detection sensitivity for capturing entire 3D image volumes with single-shot excitation.

The use of strong micron-size absorbers is an important prerequisite for successful in vivo application of the LOT technique as they effectively break the continuity in the absorption distribution induced by densely-packed RBCs and can easily be localized, even without employing broad angular coverage for OA image acquisition. Indeed, LOT is capable of rendering accurate vascular images under limited-view conditions, thus overcoming another major limitation of the commonly employed OA imaging systems[20]. Limited-view effects are associated with the speckle-free nature of OA and have been commonly averted by artificially inducing speckle grains in the images[42–44], which is equally possible in LOT even if the injected particles cannot be individually distinguished. The ability of LOT to directly measure light fluence distribution in deep tissues and anatomical co-registration based on atlas delineation of white/gray matter structures, addresses another important long-standing challenge in biomedical optics[45]. Particularly, OA image normalization with fluence is essential for achieving accurate readings of the concentration of agents or rendering physiologically-relevant blood oxygenation measurements in the presence of spectral coloring effects in deep tissues[46,47]. Fluence estimation may further be facilitated with monodisperse droplets, which can potentially be synthetized with microfluidic chips. The achievable resolution in localization-based imaging is determined by accuracy of the localized positions, which is mainly limited by the signal-to-noise ratio (SNR). As a reference, the typical SNR for individual droplets in the images after SVD filter is ~30 dB (Suppl. Fig. 13). A slightly lower SNR was attained when detecting individual microbubbles by operating the same array in a pulse-echo US mode (Suppl. Fig. 13). Note, however, that the SNR for individual droplets in the LOT images strongly decays with depth, while it is still possible to clearly identify individual microbubbles at deeper locations with US. A direct comparison between LOT and ULM is however hampered by the fact that ULM is commonly performed with other types of arrays (linear or planar) and at higher frame rates, also involving compounding of multiple transmission events to increase the SNR[48]. Other parameters, such as laser/ultrasound intensity and pulse repetition frequency, also have a major influence on the effective contrast of the images. A number of factors may affect the resolution, such as jitter in the acquisition electronics, speed of sound aberrations or the number of localized droplets. Note that the concentration of droplets further decreases with time (Suppl. Fig. 14). The jitter may have a significant effect as well[19,49], which is however minimized by the tracking approach.

The microdroplets employed in this work have a similar size to food and drug administration (FDA)-approved microbubbles used as US contrast agents. Their demonstrated biosafety may thus foster clinical translation of LOT. Note that FDA-approved indocyanine green (ICG) optical contrast agent has a similar extinction coefficient as the IR780 dye used in our study, making it an ideal candidate for undergoing the regulatory approval process once properly encapsulated into a microparticulate agent. Beyond their relevance for LOT, strongly absorbing microparticles smaller than RBCs may emerge as powerful agents for enhancing OA contrast significantly outperforming the existing OA contrast agents based on small molecule dyes and nanoparticles. The flow of microdroplets is arguably facilitated by the low viscosity of DCM. Note that capillary stalls (not observed in our sequences) are known to occur with RBCs. Quantification of the frequency of these events may provide better insights into capillary flow and enable assessing how microparticles affect microcirculation. Note that uncertainties in the estimated velocity may also result from the fact that microdroplets are found in different capillary segments in consecutive frames, particularly considering the high density of capillaries, their small size and low blood flow velocity. Accurate tracking is essential for averting these uncertainties.

LOT can be used to promote our understanding of brain microvascular organization and function[50]. It can enhance the capabilities of OA to provide hemodynamic readings e.g. of oxygen saturation or cerebral blood volume by additionally enabling the quantification of cerebral blood flow or size alteration in microvascular structures. LOT may then emerge as a unique tool to facilitate studies into neurovascular coupling and responses to stimuli, e.g. in subcortical regions not accessible with optical microscopy[51]. Cerebrovascular structural alterations and dysfunction play critical roles and are key disease biomarkers in multiple diseases such as hypertension, ischemic stroke, brain cancer, traumatic brain injury or Alzheimer's and tauopathy diseases[52–54]. Magnetic resonance imaging (MRI) and x-ray computed tomography (CT) have been employed to detect vessel remodeling, blood flow disturbances and vascular density in humans and in murine disease models, yet those methods provide sub-optimal spatial resolution and contrast for studying microvascular alterations in a living murine brain[55]. Two-photon microscopy and LSCI have alternatively been used to map the microvascular blood flow, but are only applicable at shallow depths (<1 mm from the accessible surface)[26,31,56].

In the murine model of ischemic stroke, we combined LOT with spectroscopic OA to reveal statistically significant differences in microvascular density, blood flow velocity and oxygen saturation between the ipsi- and contra-lateral sides. The 2-3 mm depth of the penetrating vessels observed noninvasively with LOT is in congruence with observations with light sheet microscopy made in cleared brains[57], further scaling well with the vessel depth previously observed in the rat brains by ULM[17]. The ability of LOT to visualize blood flow dynamics in the middle and anterior cerebral arteries (both pial and penetrating arterioles) is key for studying the mechanisms of reperfusion after middle cerebral artery occlusion[58], monitoring treatments for protecting neuroplasticity after stroke or characterizing intracerebral hemorrhages following immunotherapy[59]. The successful multi-parametric and multi-scale characterization of stroke in a murine model demonstrated in this work evinces the advantages of combining the super-resolution imaging capability of LOT with its rich functional and molecular optical contrast, the latter not readily available to other localization-based modalities such as ULM. On the other hand, ULM can compensate for the limited penetration of LOT, thus contributing to a highly synergistic and complementary value of the two modalities for interrogating brain functions in health and disease.

Taken together, these enabling features can massively impact our understanding on the structural and functional properties of cerebral microvasculature under physiological and diseased conditions. In a broader perspective, LOT can be used to facilitate early diagnosis based on bio-markers associated with microcirculatory alterations in diabetes, cancer, cardiovascular disorders, ischemic stroke or neurodegenerative diseases, while additionally providing new insights into disease progression, efficacy of drugs and other therapeutic

interventions. In conclusion, the newly developed capacity for rapid volumetric mapping and characterization of microvascular structures in vivo with spatial resolution beyond the acoustic diffraction barrier is poised to provide unprecedented insights into the anatomy and function of optically opaque organisms. In combination with the unique spectroscopic optical contrast and ultrafast imaging speed provided by state-of-the-art OA systems, the approach can massively impact a large number of studies into clinically relevant cardiovascular conditions thus further foster the growing use of OA in biology and medicine.

## Methods

### Microdroplet synthesis and characterization

A suspension of dichloromethane (DCM, Sigma Aldrich, 270997) microdroplets in water was produced following a standard emulsification procedure. The dispersed (DCM) phase was prepared by diluting 30 g of IR-780 iodide (Sigma-Aldrich, 425311) in 200 ml of DCM to achieve a dye concentration of approximately 200 mM. The continuous (water) phase was prepared by adding 3% (v/v) Tween 20 surfactant (Sigma Aldrich, P1379) in phosphate-buffered saline (PBS, Sigma Aldrich, 79378). 25 µl of disperse phase and 2 ml of continuous phase were mixed in a 2 ml Eppendorf tube and vigorously vortexed at speed 9 for 30 s (Vortex Genie 2, Scientific Instruments). The continuous phase was cooled down to approximately 4 °C before vortexing to avoid vaporization of DCM. The resulting emulsion was eventually filtered with a cell strainer with pore size 10 µm (PluriStrainer 10 µm, pluriSelect Life Science, Leipzig, Germany) and subsequently inspected in a stereo microscope (Carl Zeiss AG, Oberkochen, Germany) to verify that no particles with irregular (non-spherical) shape, arguably corresponding to the dye powder, are present in the suspension. The diameters of the droplets obtained after filtering the suspension with the cell strainer with pore size 10 µm were measured in the images taken with a bright field microscope (Leica Camera AG, Wetzlar, Germany, ×5 objective). Specifically, a total of 500 microdroplets were automatically detected and characterized (MATLAB R2019a function imfindcircles) and the results were displayed as histograms (Fig. 1b).

### Optoacoustic imaging system

OA imaging of the mouse brain was performed with a tomographic system schematically illustrated in Fig. 1a. A spherical array of 512 US sensing elements with 40 mm radius and total angular aperture of 150° (Imasonic SaS, Voray, France) was used to collect the corresponding OA signals generated via excitation with an optical parametric oscillator (OPO)-based short-pulsed (<10 ns) laser (Innolas GmbH, Krailling, Germany) tuned to 780 nm, corresponding to the peak absorption of the dye. The elements of the US array have trapezoidal shape with approximate dimensions of $3.3 \times 3.8 \, mm^2$, 7 MHz central frequency and >80% detection bandwidth. These elements are uniformly distributed over 13 rings lying on the spherical surface. The array has an 8 mm diameter cylindrical aperture in its center and three additional 4 mm diameter lateral apertures located at 45° elevation angle and equally spaced (120°) in the azimuthal direction. Light was guided with a custom-made 4-arm fiber bundle (CeramOptec GmbH, Bonn, Germany) through the array's apertures. This provided an approximately uniform illumination profile on the mouse brain surface with optical fluence <20 mJ/cm². The OA signals were amplified by ~40 dB and digitized at 40 megasamples per second with a custom-made data acquisition system (DAQ, Falkenstein Mikrosysteme GmbH, Taufkirchen, Germany) triggered with the Q-switch output of the laser and transmitted to a computer via Ethernet.

### Animal models

All animals used in this study were housed in ventilated cages inside a temperature-controlled room under a 12-h dark/light cycle. The temperature was 21 ± 1 °C, with a relative humidity of 55 ± 10%. Pelleted food and water were provided ad libitum. All experiments except for the biosafety study were performed in accordance with the Swiss Federal Act on Animal Protection and were approved by the Cantonal Veterinary Office Zürich. The biosafety study was done in accordance with Spanish and European regulations and approved by the Xunta de Galicia.

### In vivo LOT imaging

Female athymic nude-Fox1nu mice ($n = 5$, 6–8 weeks old, Janvier Lab, France) were used for in vivo imaging of healthy cerebral microcirculation. Mice were anesthetized with isoflurane (4% v/v for induction and 1.5% during the experiments, Abbott, Cham, Switzerland) in an oxygen/air mixture (100/400 ml/min). OA imaging of the brain region was performed with the head of the mouse fixed into a customdesigned stereotactic mouse head holder coupled to a breathing mask (Narishige International, Japan). Blood oxygen saturation, heart rate and body temperature were continuously monitored (PhysioSuite, Kent Scientific) and the temperature was maintained at ~36 °C with a heating pad. Bolus injection (i.v.) of 100 µl of the microdroplet emulsion was performed 30 s after the beginning of data acquisition, where the total acquisition time was 420 s. The mice were euthanized under deep anesthesia (5% isoflurane for 5 min) and subsequently decapitated without waking them up.

### Stroke model

Acute ischemic stroke was induced in both male and female BALB/c mice ($n = 5$, 8–12 weeks old, Charles Rivers, no. 028), weighting between 20 and 30 g, as described previously[39,60]. Briefly, the mice were anesthetized with 4% isoflurane in 100% $O_2$ for stroke induction. A glass pipette (calibrated at 15 mm/µl; Assistant ref. 555/5; Hoechst, Sondheim-Rhoen, Germany) was introduced into the lumen of the middle cerebral artery (MCA) and 1.5 µl of purified human alpha-thrombin (1 µl; HCT-0020, Haematologic Technologies Inc., USA) was injected to induce the formation of a clot in situ. The pipette was removed 10 min after thrombin injection. Cerebral blood flow (CBF) was monitored using LSCI (FLPI, Moor Instruments, United Kingdom). Ischemia induction was considered stable when CBF dropped to at least 50% of baseline level in the MCA territory[61]. Ischemia induction was considered stable when CBF dropped to at least 50% of baseline levels in the MCA territory. All operated animals showed stable ischemia induction and no mice were excluded.

### Image reconstruction and processing

OA images for a volume of $10 \times 10 \times 5 \, mm^3$ (100×100×50 voxels) were reconstructed with a graphics processing unit (GPU)-based implementation of a back-projection formula[62]. Prior to reconstruction, the collected signals were band-pass filtered between 0.1 and 9 MHz. A singular value decomposition (SVD) clutter filter was further applied to the raw OA signal data to reconstruct the OA images used for single droplet localization[63]. Specifically, the acquired data was divided into sets of 500 frames and a threshold on the singular vectors of the data space-time matrix corresponding to these frames was established. For each set, the first 20 and the last 100 singular vectors were filtered out. This enabled isolating the signal fluctuations ascribed to absorbing particles flowing in the vasculature from the static (background) signal coming from endogenous absorbers such as hemoglobin. A reference OA image was also reconstructed after applying the same filter for comparison purposes (Fig. 2a). The penetrating (vertical) vessels were enhanced with a Sobel filter. The filtered image was calculated via $G = sqrt(Gx^2 + Gy^2)$, where $Gx$ and $Gy$ are the filtered LOT images after convolution with 3×3×3 Sobel edge detection kernels corresponding to gradient operators in the $x$ and $y$ directions, respectively.

## Image registration

Registration between OA images and Allen mouse brain atlas was performed to further provide a better anatomical reference for identifying cerebral vascular structures. Specifically, an annotated Allen brain atlas[25] was used to identify brain regions in the original OA as well as in the LOT datasets. The reference atlas was manually aligned with the OA images (Amira 5.4.3) and for annotation of the vessels[64], which provided the best contrast for distinguishing the cerebral vasculature.

## Droplets localization and tracking

Isolated microdroplets are strong absorbers smaller in size than the resolution of our imaging system and thus appear in the image as the local point spread function (PSF). In each reconstructed frame, local intensity maxima were detected, and small regions around these maxima were correlated to a model PSF of our imaging system. Note that the same empirical PSF was used over the whole reconstructed volume corresponding to the OA response from a point source (Suppl. Fig. 15). The maxima with correlation coefficients above 0.5 were considered as droplets. Localization of these droplets was then further refined using a local quadratic fitting of the intensity maxima, and their positions were stored. A particle tracking algorithm was then used on these positions (simpletracker.m available on mathworks ©Jean-Yves Tinevez, 2019, wrapping matlab munkres algorithm implementation of ©Yi Cao 2009) in order to track the droplets over consecutive frames. A maximal linking distance of 0.5 mm was selected, which corresponds to a maximum particle velocity of 50 mm/s for the imaging frame rate.

## Fluence estimation

The amplitude of the OA signal generated by a droplet is proportional to the amount of energy absorbed times the local light fluence. The absorbed energy depends on the volume of the droplet, which changes according to the size distribution. However, given the narrow distribution of the droplet size, the light fluence can be assumed to be proportional to the average intensity of the OA signals reconstructed for droplets at a given location. For fluence estimation, 50 droplets were selected in a cross-sectional image of the brain. The intensities of the differential OA images for the positions of the selected droplets were fitted to an exponentially decaying function. Specifically, a function of the form $(a/z)\exp(-bz)$ was used for least square fitting, being $z = \text{sqrt}((x-x0)^2 + (y-y0)^2))$. Curve fitting resulted in optimum values of the parameters $a$, $b$, x0, and y0.

## Head-post and cranial window implantation for two-photon imaging

The cleaned skull of male and female BALB/c mice ($n = 3$, 8–12 weeks old, Charles Rivers, no. 028) was coated with a bonding substance (Gluma Comfort Bond; Heraeus Kulzer), which was then polymerized using a portable blue light source (600 mW/cm²; Demetron LC). Dental cement (EvoFlow; Ivoclar Vivadent AG) was used to attach a custom-made aluminum head post to the bonding agent for a stable and repeatable fixation in the microscope setup. Antibiotic ointment (Neomycin, Cicatrex; Janssen-Cilag AG) was used to treat the skin lesion, and acrylic glue (Histoacryl, B. Braun) was used to close it. Animals were given analgesics (buprenorphine 0.1 mg/kg bodyweight; Sintetica) and kept warm after surgery. Using a dental drill, a 4×4 mm craniotomy was made above the somatosensory cortex (centered above the left somatosensory cortex 3×3 mm from Bregma and 3.5×4 mm lateral). The exposed dura mater was covered with a square coverslip (3×3 mm, UQG Optics), which was cemented to the skull using dental cement.

## Two-photon imaging

BALB/c mice ($n = 3$, 8–12 weeks old, Charles Rivers, no. 028) mice were given two weeks to recover after cranial window implantation before two-photon imaging. A custom-built two-photon laser scanning microscope with a 25x water immersion objective (W-Plan-Apochromat 25x/1.0 NA, Olympus) was used for imaging equipped with a tunable pulsed laser (Chameleon Discovery TPC, Coherent Inc). Mice were anesthetized with isoflurane (4% induction, 1.5% maintenance) and head-fixed in the two-photon microscope. Texas Red dextran (5% w/v, 70,000 kDa mw, 50 ml, Life Technologies catalog number D-1864) was administered intravenously through the tail vein to facilitate visualization of vascular structures. GaAsP photomultiplier modules (Hamamatsu Photonics) equipped with 475/64, 520/50, and 607/70 band pass filters and separated by 506, 560, and 652 nm dichroic mirrors were used to detect the fluorescence emission (BrightLine; Semrock). The microscope was operated with a customized version of ScanImage (r3.8.1; Janelia Research Campus; Pologruto et al.[65]). Z stacks ($x$, $y$, and $z$ images) were captured with 1 μm step size, 512×512 pixels, and 0.74 Hz in the appropriate regions, generally spanning a vascular volume of 240x240x300 μm³. After baseline acquisition, 100 μl of DCM microbubble suspension was infused as a bolus via tail vein injection. The stacks from the same regions of interest used for baseline measures were acquired again after the DCM bubble infusion.

The acquired TIF files were analyzed with the ImageJ software (NIH, version 1.41). The z-stacks were sliced in steps of 20 μm and total number of capillaries (vessels smaller than 10 μm diameter) as well as the total number of droplets were counted in all slices. DCM stalls were counted and divided by the total amount of capillaries and droplets per z-stack, to obtain the percent of stalled capillaries and arrested droplets in each stack. In no cases did we observe arteries, arterioles, veins, or any larger vessel than a capillary (>-5–6 μm) obstructed by one or several DCM bubbles.

## Laser speckle contrast imaging

Cortical perfusion was assessed with a laser speckle contrast imaging (LSCI) monitoring system (FLPI, Moor Instruments, UK) before, during and after DCM injection in BALB/c mice ($n = 3$, 8–12 weeks old, Charles Rivers, no. 028). A baseline of 5 min was acquired and then 100 μl of DCM microbubble solution was injected as a bolus intravenously. Imaging was continuously performed for 10 minutes after DCM injection. Images are generated with arbitrary units in a 16-color palette by the MoorFLPI software V3.0.

## Biosafety study

The biosafety of the microdroplets was assessed in female Swiss mice ($n = 12$, 7 weeks old) injected i.v. with a single dose of 100 μl of (1) PBS ($n = 4$, control group), (2) DCM microdroplets ($n = 4$) and (3) DCM microdroplets containing IR780 ($n = 4$). Animals were randomly assigned to the three groups. Animals were subjected to a clinical surveillance protocol for two weeks, with regular weight control (3 times a week). Blood samples were obtained, and hematological analysis was performed by using a Mindray BC5000-Vet analyzer at days 1, 7, and 14 post-injection. At the end point (day 14 post-injection), animals were sacrificed and tissues for histopathology and serum samples for clinical biochemistry analysis were collected. Clinical biochemistry was performed in a VetScan VS2 analyzer (Zoetis) by using a Comprehensive Diagnostic Profile. Selected tissues (heart, lung, liver, kidneys, and spleen) were formalin fixed, paraffin embedded and subjected to histopathological analysis on H&E-stained sections.

## Reporting summary

Further information on research design is available in the Nature Portfolio Reporting Summary linked to this article.

## Data availability

The raw datasets before image reconstruction are too large to be publicly shared, yet they are available for research purposes from the corresponding author upon request. The entire Allen Mouse Brain

Atlas dataset and associated tools are available through an unrestricted web-based viewing application (http://mouse.brain-map.org).

## Code availability
The algorithms used for data processing are described and referenced in the manuscript. The specific implementation of the code that supports the findings of this study is available for research purposes from the corresponding author upon request.

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

## Acknowledgements

X.L.D.-B. acknowledges support from Innosuisse (application no. 51767.1 IP-LS) and the Helmut Horten Stiftung (Project Deep Skin). D.R. acknowledges support from the Swiss National Science Foundation (310030_192757) and the US National Institutes of Health (R01-NS126102-01). S.W. acknowledges support from the Swiss National Science Foundation (PP00P3_202663 and 310030_200703), the UZH Clinical Research Priority Program Stroke and the Swiss Heart Foundation.

## Author contributions

X.L.D.-B. conceived the idea of the project, X.L.D.-B. and D.R. designed and supervised the study, X.L.D.-B., D.N., C.G., J.D., J.S.-L., Z.C., Q.Z., and M.E.A. performed the experiments, X.L.D.-B., J.R., R.N., J.Z., C.G., J.D., Q.Z., A.V., M.A., and M.E.A. analyzed the data. B.W., S.W., A.V., M.A., M.E.A., and D.R. supervised the findings of the study. All authors discussed the results and contributed to writing the final manuscript.

## Competing interests

The authors declare no competing interests.
