## [Peer Review File · Nature Communications]

Deep optoacoustic localization microangiography of ischemic stroke in miceREVIEWER COMMENTS

Reviewer #1 (Remarks to the Author):

The new manuscript by Dean-ben and his colleagues has demonstrated an exciting application of 3D photoacoustic tomography or optoacoustic tomography, which has achieved super-resolution tracking of highly-absorbing NIR droplets in 3D mouse brain. The spatial resolution has been improved to 20 μm in small animal brain imaging, with the scalp and skull intact. More importantly, the hemodynamic information such as blood flow and blood oxygenation have been quantified with improved accuracy at microvessel resolution, which has never been achieved before in optoacoustic imaging. This exciting new technology has demonstrated its high potential in small animal brain imaging by mapping the microvascular response in an ischemic stroke mode. Overall, I think this is a highly exciting work done by a reputable group in optoacoustic imaging. The manuscript is well written with sound technical details. The results are convincing. I highly recommend this manuscript for publication in Nature communications. Below are my comments and suggestions mainly about the technical aspects of this work that might help improve the manuscript.

1. The novelty of this work is clearly described by the authors, particularly compared with the previously published 2D photoacoustic tomography of highly-absorbing droplets and cells (ref 22 and 23). 3D tracking is a significant technical advance over 2D tracking. Nevertheless, the technical advantage of LOT over the well-established localization ultrasound microscopy (LUM) is not as clear. Since both technologies use exogenous contrast, LUM benefits from the clinically approved bubbles while LOT still has not been clinically approved. Moreover, the spatial resolution, imaging depth, and the flow speed quantification achieved by LOT in this work seems less striking than the published LUM work. Thus, this manuscript can benefit from more information about the technical benefit of LOT over LUM.

2. Following the last question, the imaging depth achieved by LOT is only 3 mm from the skin surface, which is somewhat low for this imaging system with 7 MHz detection frequency and 780 nm excitation wavelength. I would expect a penetration depth on the level of 10 mm which should penetrate through the whole-brain with scalp and skull intact. Is this because the droplet signal is too weak compared with the blood signals at deeper brain? If this is true, then the practical advantage of LOT over other pure optical imaging techniques or LUM is less clear. Any additional data about the true imaging depth with and without the particle tracking?

3. The highly-absorbing droplets is innovative given that its absorption is 10,000 times higher than single RBCs. The authors have a thorough discussion on the optical properties of the droplets. One question is that given the high concentration of the dye within the droplet, what is the effective penetration depth of light into the droplet? Will this effective penetration depth impact the actual PA signals generated by the droplet and thus affect the light fluence estimation?

4. Another question about the droplet is its biosafety. It is understood that the dye is encapsulated while flowing inside blood vessels. However, it is anticipated that droplets may break and release the high-concentration dye into the blood stream. Will this be a safety concern? Have the authors studied the clearance rate of the droplets? Or do they observe any free dye signals in the blood during the experiments?

5. The light fluence estimation is extremely important for the quantification of blood oxygenation. LOT can provide a quantitative map of the light fluence. My question is how much improvement has been achieved by applying the fluence map into the oxygenation calculation? Can the authors show the oxygenation results with and without the fluence compensation?

6. Following the question above, in the stroke mode, as the blood perfusion is impaired or even stopped in the stroke region, the droplets are expected to be less frequent or even absent in this region. If this is the case, how is the blood flow estimated in these vessels? And how does this impact the light fluence quantification if there are fewer or no droplets?

7. How much time is needed to track sufficient events of the droplets in the demonstrated experiments? What is the maximum flow speed that can be tracked given the relatively low pulse rep rate of the laser? How is LOT compared with LUM in terms of imaging speed (frame rate)?

Junjie Yao (Duke University)

Reviewer #2 (Remarks to the Author):

In this manuscript, Deán-Ben et al. report an improved method for in-vivo volumetric optoacoustic (OA) imaging of deep tissue microvasculature. The paper first describes the current capabilities of OA imaging, which is limited in depth by frequency-dependent acoustic attenuation to about ~1 mm deep in the tissue. Localization optoacoustic tomography (LOT) can be used to detect individual circulating microbubbles and form a higher resolution image of the vasculature by localizing them across a series of images. Measurement of blood flow velocities through single-particle tracking is also possible, though the requirement that individual absorbing particles be sparsely concentrated in the blood stream makes the background absorption of red blood cells an important limiting factor. To improve OA imaging with localization, Deán-Ben et al. generated highly absorbing dichloromethane (DCM) microdroplets similar in size to red blood cells and demonstrated volumetric OA imaging with single-particle detection after filtering and image registration, with no temporal integration required. Continuous flows of particles could be observed, demonstrating high-resolution particle tracking across a depth range of over 3 mm through an intact scalp and skull. Finally, using a stroke model, the paper demonstrates mapping microvascular density, blood flow speed, and oxygenation in brain regions both ipsilateral and contralateral to the stroke, finding clear reductions in vessel perfusion, flow speed, and oxygenation in the brain ipsilateral to the stroke relative to the contralateral region.

Overall, this work demonstrates a substantial improvement in OA imaging and would be a highly useful method for mapping blood flow dynamics in vivo. There are some concerns and questions about this work, though if these are addressed, this paper would be appropriate for publication in Nature Communications.

Larger concerns/questions:

1. In Reeson, Choi, and Brown (eLife, 2018), injection of 4 μm diameter microspheres (2% solids in 20 μL) led to blockage of ~4% of cortical capillaries. Thus, the particle size being less than red blood cell size is not sufficient to infer that capillary occlusions do not occur. Other factors about the particles, such as surface characteristics that influence interaction with the endothelium, may contribute. In this manuscript, injected microdroplets had a mean diameter of 5.5 μm with some as large as ~10 μm (Fig 1b). While it is true that the deformability of the DCA droplets may play some role in reducing their tendency to obstruct vessels and the overall concentration of droplets is relatively low, it still seems like this is an important issue to address well. Could the authors characterize the density of arrested droplets, perhaps histologically? What fraction of capillaries are occluded? What impact on overall cerebral blood flow would this have?
2. Are baseline measurements of flow speed before the stroke available? Characterizing flow changes in some of the large cerebral vessels after the MCA occlusion would be of interest. For example, flow speed increases in the ipsilateral PCA and ACA are expected, as well as flow reversals along the leptomeningeal anastomoses.
3. The spatial filtering of the low resolution red blood cell oxygenation data with the higher resolution vascular map is suspect. Wouldn't the low resolution of the oxygenation data lead to "contamination" of the oxygenation measurement in one vessel by nearby vessels. Spatially filtering would not eliminate this cross talk, leading to errors in the oxygenation estimate for individual vessels.
4. Why were only female mice used in this baseline study?
5. For the stroke model, how many animals were excluded for not exhibiting stable ischemic induction?

Smaller questions/comments/typos:

1. For image processing methods, what software was used for filtering, thresholding, etc.?
2. Line 334 & 335: "8 to 12 weeks" is repeated.
3. Line 389: Light fluence?
4. Line 559 & 561: Inset or inlet?
5. Figures 1b, 2b, 2e: Axis arrows are barely visible.
6. Figure 1d: Inconsistent color scheme?

In summary, the generation of highly absorbing DCM particles and volumetric localization OA imaging

demonstrated by Deán-Ben et al. represents a significant improvement over current methods, and the application to mapping blood flow speed and oxygenation in deep tissues after stroke is a nice demonstration application. With the above concerns and questions addressed, I recommend publication in Nature Communications.

Reviewer #3 (Remarks to the Author):

In the manuscript the authors present a new imaging approach based on combined optoacoustic imaging (OA) and localization optoacoustic tomography (LOT) finalized at providing microangiography of ischemic strokes in mice. This is performed at outstanding speed and resolution, using microdroplets for obtaining contrast in LOT.

The possibility to combine both functional (spectroscopic OA) and structural (LOT) data is strikingly important because currently there is no imaging tool that allows that. Also highly important is the fact that the method is non invasive and does not require any window implantation making it easily implementable. Also the resolution (20 microns) achieved at 2-3 mm depth is extremely impressive and groundbreaking in the presented context.

My overall assessment from a technical imaging technique is that the manuscript represents an extremely exciting contribution to the optical imaging literature. While the method has been used in mice it could be hopefully extended to other models. But overall I can see a great use of this methodology in several imaging settings.

The manuscript is extremely well written with a large amount of carefully given details, and very clear in its content and description.

The exposition is very fluid and clear as also the methods and conclusions. Also the language is pretty fluent.

The data are very nicely collected and the images appear to be impeccable and of top quality. I was greatly impressed by them and they set a standard for future work in the presented context.

I would recommend in Fig. 3 to present equivalent data, as obtained for the 3 stroke mice, for a control one.

Also in Movie 1, I would recommend a movie where MIP projections of the signals along the three axial components is presented. This perhaps will emphasize more the correlative position of the two signals.

Another question I have regards the microparticles used in the experiment. Has their toxicity be determined?

Point-by-point response to the Reviewers' comments

We thank the three Reviewers for the valuable comments and suggestions. We have performed new experiments and simulations to address all the concerns raised.

Reviewer #1

The new manuscript by Dean-ben and his colleagues has demonstrated an exciting application of 3D photoacoustic tomography or optoacoustic tomography, which has achieved super-resolution tracking of highly-absorbing NIR droplets in 3D mouse brain. The spatial resolution has been improved to 20 um in small animal brain imaging, with the scalp and skull intact. More importantly, the hemodynamic information such as blood flow and blood oxygenation have been quantified with improved accuracy at microvessel resolution, which has never been achieved before in optoacoustic imaging. This exciting new technology has demonstrated its high potential in small animal brain imaging by mapping the microvascular response in an ischemic stroke mode. Overall, I think this is a highly exciting work done by a reputable group in optoacoustic imaging. The manuscript is well written with sound technical details. The results are convincing. I highly recommend this manuscript for publication in Nature communications. Below are my comments and suggestions mainly about the technical aspects of this work that might help improve the manuscript.

Reply: We thank the Reviewer for the positive comments and recommendation for publication of the manuscript. Please find below the answers to the specific questions.

1. The novelty of this work is clearly described by the authors, particularly compared with the previously published 2D photoacoustic tomography of highly-absorbing droplets and cells (ref 22 and 23). 3D tracking is a significant technical advance over 2D tracking. Nevertheless, the technical advantage of LOT over the well-established localization ultrasound microscopy (LUM) is not as clear. Since both technologies use exogenous contrast, LUM benefits from the clinically approved bubbles while LOT still has not been clinically approved. Moreover, the spatial resolution, imaging depth, and the flow speed quantification achieved by LOT in this work seems less striking than the published LUM work. Thus, this manuscript can benefit from more information about the technical benefit of LOT over LUM.

Reply: We thank the Reviewer for this insightful comment. Indeed, LUM provides super-resolution imaging of microvascular structures based on a clinically-approved contrast agent (microbubbles), which represents an important advantage. Note, however, that optoacoustic imaging generally provides very different (complementary) functional and molecular information invisible with ultrasound. Indeed, we have shown that LOT can map microvascular and oxygen saturation alterations in a murine model of stroke. Thereby, we believe that LUM and LOT can complement each other rather than replace each other. We elaborated on these complementary advantages of each approach in the revised manuscript (page 6, paragraph 3 and page 7, paragraph 4). On the other hand, we have performed a new experiment to facilitate a comparison of the technical performance of LUM and LOT, particularly in terms of spatial resolution and imaging depth. For this, OA and US imaging and tracking of intravenously-injected microdroplets and microbubbles, respectively, was performed as they flow through the mouse brain vasculature. The results of this experiment are included as supplementary information (Suppl. Fig. 12). The resolution in localization imaging is mainly determined by the signal-to-noise ratio (SNR) of individual particles in the images, namely, droplets for LOT or microbubbles for LUM. We found that the SNR of the

droplets imaged with OA in the cortical surface was approximately 5 dB higher than that for microbubbles imaged with US. However, a strong attenuation was observed in the droplet signals for depths beyond 2 mm, while microbubbles at larger depths could clearly be observed with US. This indicates that LOT can achieve higher resolution at shallow regions, but the SNR degrades faster with depth than for LUM. These results are now discussed in the revised manuscript, also considering the technical limitations of such comparison (page 6, paragraph 2). Note that LUM is commonly performed with another type of ultrasound arrays (typically planar) and at higher imaging rates, which are generally not suitable for OA tomographic imaging. The driving voltage of the transducers and pressure levels at the bubble locations is also an important factor to consider. Regarding the capabilities of LOT to image blood flow, we have shown that it is possible to capture velocities of up to 50 mm/s in major vessels and slower velocities in microvascular networks. Therefore, LOT can cover the typical range of blood flow velocities in vivo when employing volumetric image acquisition at 100 Hz pulse repetition frequency and this can be increased if the appropriate laser source is available. Note also that one major benefit of LOT is its simultaneous ability to provide oxygen saturation maps (as shown in Fig. 3), which is impossible to attain with LUM.

2. Following the last question, the imaging depth achieved by LOT is only 3 mm from the skin surface, which is somewhat low for this imaging system with 7 MHz detection frequency and 780 nm excitation wavelength. I would expect a penetration depth on the level of 10 mm which should penetrate through the whole-brain with scalp and skull intact. Is this because the droplet signal is too weak compared with the blood signals at deeper brain? If this is true, then the practical advantage of LOT over other pure optical imaging techniques or LUM is less clear. Any additional data about the true imaging depth with and without the particle tracking?

Reply: Indeed, the signals from the droplets are too weak to be detected for deeper brain regions. We ascribe this to the strong light attenuation in the mouse brain, which also affects standard optoacoustic images acquired with the same array. Indeed, the microdroplets are only detectable in regions where blood vessels can be clearly visualized. As mentioned in the answer to the previous comment, we have performed a new experiment to compare the achievable depth with LOT and LUM. We have shown that the SNR of the droplets in the singular value decomposition (SVD)-filtered LOT images decays by ~ 7 dB at depths of 3-4 mm relative to the surface (Suppl. Fig. 12). The achievable depth can potentially be enhanced with other types of particles strongly absorbing at longer wavelengths (e.g. 1064 nm). Yet, with the currently achievable depth of ~ 3 mm the LOT approach offers important advantages not available with the existing modalities, in particular considering the powerful combination of high resolution with label-free extraction of blood oxygenation parameters.

3. The highly-absorbing droplets is innovative given that its absorption is 10,000 times higher than single RBCs. The authors have a thorough discussion on the optical properties of the droplets. One question is that given the high concentration of the dye within the droplet, what is the effective penetration depth of light into the droplet? Will this effective penetration depth impact the actual PA signals generated by the droplet and thus affect the light fluence estimation?

Reply: We thank the Reviewer for this important question. We provide an estimation of the total absorbed energy in the droplets accounting for the strong light attenuation in the supplementary information of the revised manuscript (Suppl. Note 1). Note that the amount of absorbed energy still depends on the fluence. The estimation of light fluence is more affected by the fact that the microdroplets are

polydisperse (multiple sizes) rather than by light attenuation. However, the fluence can still be estimated by fitting a curve to the signals measured from a relatively large number of droplets. We performed numerical simulations to validate the accuracy of this approach in Suppl. Note 2. We highlight this issue in the revised manuscript (page 4, paragraph 2).

4. Another question about the droplet is its biosafety. It is understood that the dye is encapsulated while flowing inside blood vessels. However, it is anticipated that droplets may break and release the high-concentration dye into the blood stream. Will this be a safety concern? Have the authors studied the clearance rate of the droplets? Or do they observe any free dye signals in the blood during the experiments?

Reply: This is an important aspect. With the help of a toxicology expert M. Arand, now coauthoring the manuscript, we have assessed the potential toxic effects of the dichloromethane (DCM) microdroplets at the concentration levels used in our study. The injected amount of DCM, mainly present in sparsely distributed droplets, was determined to be below safety standards in humans for ~16 hour exposure in the work environment (Suppl. Note 3). Following the comments of all Reviewers and the recommendation of the Editor, we have further performed a toxicology study to assess the biosafety of the droplets. For this, we have performed new experiments in three groups of Swiss mice (n=4 each). These were injected with 100 μ l of 1) an emulsion of the microdroplets used in the experiments, 2) an emulsion of DCM microdroplets (without IR780 dye), and 3) phosphate buffered saline (PBS, control), respectively. The mice were weighted and scored during 14 days, hematology was performed at days 1, 7 and 14 post injection, and blood biochemistry and histology of major organs were performed at day 14 (final point). The results of this new experiment are included in a new figure (Fig. 4) and as supplementary information (Suppl. Fig. 11) in the revised manuscript. No signs of toxicity were observed. On the other hand, following the suggestion of Reviewer #2 and the editor, we have also performed new experiments to evaluate if capillary occlusion is produced. Specifically, we have performed 2-photon microscopy imaging experiments in BALB/c mice. The results of this new experiments are presented in a new figure (Fig. 4) and discussed in the revised manuscript. Overall, very rare occlusion events were observed with their incidence being comparable to occlusions produced by red blood cells and leukocytes.

5. The light fluence estimation is extremely important for the quantification of blood oxygenation. LOT can provide a quantitative map of the light fluence. My question is how much improvement has been achieved by applying the fluence map into the oxygenation calculation? Can the authors show the oxygenation results with and without the fluence compensation?

Reply: This is an excellent question. Unfortunately, light fluence estimation can only be performed for wavelengths close to the peak absorption of the dye, where the signals from the droplets are sufficiently strong. In order to improve accuracy of the oxygen saturation readings, the fluence compensation must be performed over a broader wavelength range, which may be achieved with other type of droplets or microparticles. With the currently available methodology we are unable to claim extraction of quantitative map of light fluence at multiple wavelengths with LOT, which is now mentioned in the revised manuscript (page 4, paragraph 2).

6. Following the question above, in the stroke mode, as the blood perfusion is impaired or even stopped in the stroke region, the droplets are expected to be less frequent or even absent in this region. If this is the

case, how is the blood flow estimated in these vessels? And how does this impact the light fluence quantification if there are fewer or no droplets?

Reply: Indeed, the functional microvascular density (microvascular structures where blood can flow) is significantly reduced in the stroke-affected area and blood flow velocity measurements are not possible in many of these vessels. The values displayed in Fig. 3 correspond to the mean velocity in the ipsi- and contra-lateral sides, i.e., not directly corresponding to the number of functional vessels. On the other hand, as correctly pointed out by the Reviewer, light fluence may not be accurate in these regions if the number of localized droplets is insufficient. We comment on this issue in the revised manuscript (page 4, paragraph 3).

7. How much time is needed to track sufficient events of the droplets in the demonstrated experiments? What is the maximum flow speed that can be tracked given the relatively low pulse rep rate of the laser? How is LOT compared with LUM in terms of imaging speed (frame rate)?

Reply: We evaluated the LOT performance as a function of particles/frames (Suppl. Fig. 2). LOT images could be reconstructed with less than 500 frames, i.e., within 5 seconds. We emphasize this issue in the revised manuscript (page 4, paragraph 1). The time required to form an image depends also on the concentration of droplets. A higher concentration can enhance the temporal resolution but potential toxic effects must be characterized. On the other hand, as mentioned in the answer to the first comment above, we have shown that it is possible to capture velocities of up to 50 mm/s in major vessels and slower velocities in microvascular networks. Therefore, typical range of blood flow velocities in vivo can be covered with the 100 Hz pulse repetition frequency and this can be increased if the appropriate laser source is available. The frame rate can also be increased in LUM, but compounding is generally needed to achieve sufficient SNR. We refer to our answer to the first comment above.

Reviewer #2

In this manuscript, Deán-Ben et al. report an improved method for in-vivo volumetric optoacoustic (OA) imaging of deep tissue microvasculature. The paper first describes the current capabilities of OA imaging, which is limited in depth by frequency-dependent acoustic attenuation to about ~1 mm deep in the tissue. Localization optoacoustic tomography (LOT) can be used to detect individual circulating microbubbles and form a higher resolution image of the vasculature by localizing them across a series of images. Measurement of blood flow velocities through single-particle tracking is also possible, though the requirement that individual absorbing particles be sparsely concentrated in the blood stream makes the background absorption of red blood cells an important limiting factor. To improve OA imaging with localization, Deán-Ben et al. generated highly absorbing dichloromethane (DCM) microdroplets similar in size to red blood cells and demonstrated volumetric OA imaging with single-particle detection after filtering and image registration, with no temporal integration required. Continuous flows of particles could be observed, demonstrating high-resolution particle tracking across a depth range of over 3 mm through an intact scalp and skull. Finally, using a stroke model, the paper demonstrates mapping microvascular density, blood flow speed, and oxygenation in brain regions both ipsilateral and contralateral to the stroke, finding clear reductions in vessel perfusion, flow speed, and oxygenation in the brain ipsilateral to the stroke relative to the contralateral region.

Overall, this work demonstrates a substantial improvement in OA imaging and would be a highly useful method for mapping blood flow dynamics in vivo. There are some concerns and questions about this work, though if these are addressed, this paper would be appropriate for publication in Nature Communications.

Reply: We thank the Reviewer for appreciating the benefits of our approach. Please find below the specific answers to the concerns and questions raised.

Larger concerns/questions:

1. In Reeson, Choi, and Brown (eLife, 2018), injection of 4 μm diameter microspheres (2% solids in 20 μL) led to blockage of ~4% of cortical capillaries. Thus, the particle size being less than red blood cell size is not sufficient to infer that capillary occlusions do not occur. Other factors about the particles, such as surface characteristics that influence interaction with the endothelium, may contribute. In this manuscript, injected microdroplets had a mean diameter of 5.5 μm with some as large as ~10 μm (Fig 1b). While it is true that the deformability of the DCA droplets may play some role in reducing their tendency to obstruct vessels and the overall concentration of droplets is relatively low, it still seems like this is an important issue to address well. Could the authors characterize the density of arrested droplets, perhaps histologically? What fraction of capillaries are occluded? What impact on overall cerebral blood flow would this have?

Reply: We thank the Reviewer for this insightful comment. Note that visualization of the flow of particles is the basis of LOT imaging, i.e., the acquired sequence of images enables estimating whether particle arrest is produced in the imaged region. To better assess capillary occlusion, we have performed new experiments where the brains of 3 mice were imaged with 2-photon microscopy. We imaged cortical vascular networks (up to ~400 μm depth) in mice before and after injection of the microdroplets. Texas red dextran was injected intravenously to visualize the brain vessels and we identified flowing versus non-flowing capillaries by the presence or absence of streaking RBCs. Unbiased sampling of cortical vasculature in 5241 capillaries for 2 hours revealed that capillary obstructions were relatively rare after injection, affecting 10 in 2849 capillaries (~0.35%) compared to baseline measurements (2 in 2392 capillaries, ~0.083%). As also mentioned by the Reviewer, we believe that the reduced percentage of capillary

obstructions may be due to the low number of injected droplets as well as their high deformability that allows free passage through the capillary network. The results of these experiments are included and discussed in the revised manuscript (new Fig. 4, Suppl. Fig. 11, and page 5, paragraph 2). On the other hand, we also characterized the blood flow in the brain with laser speckle imaging (LSI) for mice intravenously injected with the emulsion of microdroplets used in the LOT imaging experiments and with a vehicle (phosphate buffered saline (PBS)). We found that the blood flow is reduced when the emulsion of microdroplets is injected, which can affect the accuracy of blood flow measurements. Note, however, that clear changes in blood flow velocity could still be quantified in the stroke model. We comment on this issue in the revised manuscript (Suppl. Fig. 10, and page 5, paragraph 2).

2. Are baseline measurements of flow speed before the stroke available? Characterizing flow changes in some of the large cerebral vessels after the MCA occlusion would be of interest. For example, flow speed increases in the ipsilateral PCA and ACA are expected, as well as flow reversals along the leptomeningeal anastomoses.

Reply: As suggested, we now include new data corresponding to a mouse imaged with LOT before and after stroke (Suppl. Fig. 5). As expected, a clear change in the microvascular density was produced. However, no significant increase in blood flow velocity in ipsilateral PCA and ACA vessels was observed. We do agree that flow reversals as well as increase in flow speed in large arteries may be expected after stroke. However, this may be relevant for C57BL/6 and not BALB/c mice, which lack leptomeningeal collaterals. Further studies on C57BL/6 would be necessary to understand the hemodynamic changes in collaterals and large arteries following ischemic stroke. Moreover, note that a more accurate characterization of the blood flow within a larger region requires scanning the transducer, thus subsequently increasing the acquisition time and severity of the experiments beyond what is currently acceptable by the animal experimentation regulations.

3. The spatial filtering of the low resolution red blood cell oxygenation data with the higher resolution vascular map is suspect. Wouldn't the low resolution of the oxygenation data lead to "contamination" of the oxygenation measurement in one vessel by nearby vessels. Spatially filtering would not eliminate this cross talk, leading to errors in the oxygenation estimate for individual vessels.

Reply: Indeed, cross-talk is to be expected for vessels separated by a short distance. We performed new simulations to better quantify this effect and estimate what is the distance for which differences in oxygen saturation in neighboring absorbers (oxygen saturation levels of 70% and 100% mimicking an artery and a vein) can be resolved. It was found that cross-talk artefacts leading to errors in oxygen saturation readings starts to be produced for distances larger than the acoustic diffraction limit. These errors are increased for shorter distance but it is still possible to distinguish oxygenation levels beyond the acoustic diffraction barrier. The results of these simulations are included in the revised manuscript (page 5, paragraph 1 and Suppl. Fig. 9). Note that oxygen saturation errors are also produced by other factors e.g. spectral coloring effects corresponding to differences in light attenuation for different wavelengths.

4. Why were only female mice used in this baseline study?

Reply: We thank the Reviewer for this question. Indeed, only female mice could be used for the initial basic feasibility and performance evaluation experiments due to limitations of the animal protocol. The acute ischemic stroke experiments were done in both male and female mice of a different strain (BALB/c).

5. For the stroke model, how many animals were excluded for not exhibiting stable ischemic induction?

Reply: In the thrombin stroke model, ischemia induction is considered successful when CBF dropped to at least 50% of baseline level in the MCA territory. In our hands, the success rate in this model is >95%. Here, all animals exhibited stable ischemic induction and all mice were included. We regret for any confusion and have now clarified this in the methods section as follows: "Ischemia induction was considered stable when CBF dropped to at least 50% of baseline level in the MCA territory. All operated animals showed stable ischemia induction and no mice were excluded".

Smaller questions/comments/typos:

- 1. For image processing methods, what software was used for filtering, thresholding, etc.?*
- 2. Line 334 & 335: "8 to 12 weeks" is repeated.*
- 3. Line 389: Light fluence?*
- 4. Line 559 & 561: Inset or inlet?*
- 5. Figures 1b, 2b, 2e: Axis arrows are barely visible.*
- 6. Figure 1d: Inconsistent color scheme?*

Reply: We thank the Reviewer for noticing these errors and typos. We corrected them in the revised manuscript. Note that the color scheme in Figure 1d is the same as in Figure 1c. We have added a new colorbar to make it clear.

In summary, the generation of highly absorbing DCM particles and volumetric localization OA imaging demonstrated by Deán-Ben et al. represents a significant improvement over current methods, and the application to mapping blood flow speed and oxygenation in deep tissues after stroke is a nice demonstration application. With the above concerns and questions addressed, I recommend publication in Nature Communications.

Reply: We thank the Reviewer for the recommendation. We hope to have addressed all the concerns raised.

Reviewer #3

In the manuscript the authors present a new imaging approach based on combined optoacoustic imaging (OA) and localization optoacoustic tomography (LOT) finalized at providing microangiography of ischemic strokes in mice. This is performed at outstanding speed and resolution, using microdroplets for obtaining contrast in LOT.

The possibility to combine both functional(spectroscopic OA) and structural (LOT) data is strikingly important because currently there is no imaging tool that allows that. Also highly important is the fact that the method is non invasive and does not require any window implantation making it easily implementable. Also the resolution (20 microns) achieved at 2-3 mm depth is extremely impressive and groundbreaking in the presented context.

My overall assessment from a technical imaging technique is that the manuscript represents an extremely exciting contribution to the optical imaging literature.

While the method has been used in mice it could be hopefully extended to other models. But overall I can see a great use of this methodology in several imaging settings.

The manuscript is extremely well written with a large amount of carefully given details, and very clear in its content and description.

The exposition is very fluid and clear as also the methods and conclusions. Also the language is pretty fluent.

The data are very nicely collected and the images appear to be impeccable and of top quality. I was greatly impressed by them and they set a standard for future work in the presented context.

Reply: We thank the Reviewer for highlighting the impact of our work and for the valuable suggestions to improve it. Please find below the answers to the specific comments.

I would recommend in Fig. 3 to present equivalent data, as obtained for the 3 stroke mice, for a control one.

Reply: We thank the Reviewer for this recommendation. In the revised manuscript, we include new data corresponding to an experiment where we imaged the same brain region of a mouse with LOT before and after inducing the stroke. As expected, the microvascular density was clearly reduced in the stroke-affected area. The results of this experiment are included as supplementary information in the revised manuscript (Suppl. Fig. 5). We believe however that the images of the contralateral brain regions shown in Fig. 3 represent a better control. Indeed, apart from increasing the severity of the experiments and causing stress in the mouse potentially affecting the measurements, comparing the images before and after inducing the stroke is hampered by the fact that it is generally challenging to fix the mouse in the same position. Note that a dedicated scanning system has been devised to cover both sides of the brain in order to make a proper comparison.

Also in Movie 1, I would recommend a movie where MIP projections of the signals along the three axial components is presented. This perhaps will emphasize more the correlative position of the two signals.

Reply: We thank the Reviewer for this suggestion. In the revised manuscript, we modified Suppl. Movie 1 to also show the three MIP projections during the injection of the microdroplets.

Another question I have regards the microparticles used in the experiment. Has their toxicity be determined?

Reply: This is indeed a very important point. With the help of a toxicology expert M. Arand, now coauthoring the paper, we have assessed the potential toxic effects of dichloromethane (DCM) microdroplets. The injected concentration of DCM is actually very low as it is mainly present in sparsely distributed droplets. Indeed, the amount of DCM in the injected emulsion was determined to be below safety standards in humans for ~16 hour exposure in the work environment, as supported by respective literature and elaborated in a supplementary note in the revised manuscript (Suppl. Note 3). The concentration of the IR780 contrast agent is also below the reported damage threshold in mice. To more thoroughly demonstrate the biosafety of the droplets, we have performed new experiments in three groups of Swiss mice (n=4 each). These were injected with 100 μ l of 1) an emulsion of the microdroplets used in the experiments, 2) an emulsion of DCM microdroplets (without IR780 dye), and 3) phosphate buffered saline (PBS, control), respectively. The mice were weighted and scored during 14 days, hematology was performed at days 1, 7 and 14 post injection, and blood biochemistry and histology of major organs were performed at day 14 (final point). The results of this new experiment are included in a new figure (Fig. 4) and as supplementary information (Suppl. Fig. 11) in the revised manuscript. No signs of toxicity were observed. On the other hand, following the suggestion of Reviewer #2 and the Editor, we have also performed new experiments to evaluate if capillary occlusion is produced. Specifically, we have performed 2-photon microscopy imaging experiments in BALB/c mice. The results of this new experiments are presented in a new figure (Fig. 4) and discussed in the revised manuscript. Overall, very rare occlusion events were observed, similar to the incidence of occlusions produced by red blood cells and leukocytes.

REVIEWER COMMENTS

Reviewer #1 (Remarks to the Author):

All of my comments and questions have been adequately addressed by the authors. I appreciate their additional efforts and time. This fine manuscript in my opinion is now ready for publication.

Reviewer #2 (Remarks to the Author):

The authors have largely addressed previous concerns raised in our review, but a significant question remains.

The overall perfusion decrease after injecting the microparticles measured with laser speckle contrast analysis imaging should not be so quickly dismissed. The data shown in Supplemental Fig. 10 should include a color scale bar that allows the magnitude of perfusion change to be assessed, and these changes should be quantified and compared across mice/treatments. It is not possible to assess how severe the CBF decrease is in SFig. 10 due to the lack of a color scale bar, but if there are large decreases in perfusion due to the microparticles this calls the entire method into question. Also, note that the lack of a large increase in capillary stalling does not mean that there could not be other reasons that the microparticles impact blood flow. Quantifying the degree of flow decrease, showing this flow decrease to be acceptably modest, and making an effort to understand the cause of the flow decrease all seem essential for a paper about a method to measure blood flow.

Reviewer #3 (Remarks to the Author):

I found that the authors did address excellently the points raised by myself particularly regarding the toxicity of the contrast agent with the inclusion of a new section and the collaboration of an additional expert co-author. Also other reviewers' points are properly addressed in my opinion.

Point-by-point response to the Reviewers' comments

Reviewer #1

All of my comments and questions have been adequately addressed by the authors. I appreciate their additional efforts and time. This fine manuscript in my opinion is now ready for publication.

Reply: We thank the Reviewer for appreciating our efforts and for recommending publication of our article.

Reviewer #2

The authors have largely addressed previous concerns raised in our review, but a significant question remains.

The overall perfusion decrease after injecting the microparticles measured with laser speckle contrast analysis imaging should not be so quickly dismissed. The data shown in Supplemental Fig. 10 should include a color scale bar that allows the magnitude of perfusion change to be assessed, and these changes should be quantified and compared across mice/treatments. It is not possible to assess how severe the CBF decrease is in SFig. 10 due to the lack of a color scale bar, but if there are large decreases in perfusion due to the microparticles this calls the entire method into question. Also, note that the lack of a large increase in capillary stalling does not mean that there could not be other reasons that the microparticles impact blood flow. Quantifying the degree of flow decrease, showing this flow decrease to be acceptably modest, and making an effort to understand the cause of the flow decrease all seem essential for a paper about a method to measure blood flow.

Reply: We thank the Reviewer for raising this point. We performed new experiments to thoroughly investigate the cause of the observed reduced blood flow with laser speckle contrast. Note that the laser speckle contrast imaging (LSCI) device at our disposal operates at 790 nm wavelength, i.e., close to the absorption peak of the dye. This leads to optical excitation of the dye and reemission of light in the form of fluorescence, which is temporally incoherent and thus interferes with the speckle contrast.

To test this, we performed LSCI imaging during the injection of dichloromethane droplets not containing the dye. No significant signal changes were observed in this case, which was consistent in n=3 mice. We included these results in the new Suppl. Fig. 10 of the revised manuscript (also included as Fig. R1a below). We also included a colorbar and values of cerebral blood flow (CBF) provided by the LSCI device before and after injection of IR-780 droplets (Fig. R1a-c). Furthermore, to demonstrate that the reduced laser speckle contrast was due to the fluorescence of the dye rather than blood flow reduction, we disposed ~20 µl of the emulsion of IR-780 microdroplets over the left hemisphere without any intravascular injection. As shown in Fig. R1 below, the LSCI signal was clearly reduced in the presence of the droplets (Fig. R1c), even though the concentration of the dye is sufficiently low not to result in a visible (green) coloration. Altogether, these data show that the droplets containing the dye altered the signal of LSCI, not the brain perfusion nor blood flow.

Figure R1. Laser speckle contrast imaging (LSCI) of the mouse brain cortex. (a) LSCI image and quantification after intravenous injection of droplets not containing the dye. (b) LSCI image and quantification after intravenous injection of IR-780 droplets (c) LSCI image taken after depositing $\sim 20 \mu\text{l}$ of the emulsion of IR-780 droplets on top of the left hemisphere of the skull.

Note that LSCI does not provide quantitative values of CBF but rather changes in speckle contrast related to blood flow. Therefore, to better quantify blood flow velocity and to demonstrate that dichloromethane droplets containing the dye do not result in flow changes, two additional experiments were performed. In the first experiment, the blood flow velocity was estimated with 2 photon microscopy in several brain microvessels before and after injection of the microdroplet emulsion. In the second experiment, a high-speed camera was used to track fluorescent microbeads smaller than $5 \mu\text{m}$ (filtered with a $5 \mu\text{m}$ mesh) in the mouse cortex after removing the scalp. No significant differences were observed before and after

injection of the IR-780 microdroplets. Note that the microdroplets were not visible at the 420 nm wavelength used for exciting the microbeads. The results of these two experiments are shown in Fig. R2 below. The two figures could be included as supplementary information if requested. Based on these results, it is safe to conclude that no significant blood flow changes are induced with the microdroplet emulsion. Note however that our method is not chiefly aimed at quantifying blood flow but rather enhancing the resolution of vascular optoacoustic imaging.

Figure R2. Blood flow velocity quantification. (a) Two-photon images taken before and after injection of microdroplets. (b) Measured blood flow velocity at selected points before and after injection. No significant differences were observed ($p=0.22$ for a paired t-test). (c) Blood flow velocity maps acquired by tracking $<5 \mu\text{m}$ fluorescent microbeads in the mouse cortex. (d) Measured blood flow velocity in vessels with a velocity range from 6-18 mm/s before and after injection. No significant differences were observed ($p=0.72$ for a paired t-test).

Reviewer #3

I found that the authors did address excellently the points raised by myself particularly regarding the toxicity of the contrast agent with the inclusion of a new section and the collaboration of an additional expert co-author. Also other reviewers' points are properly addressed in my opinion.

Reply: We thank the Reviewer for the positive evaluation of the manuscript and for recommending publication.

REVIEWERS' COMMENTS

Reviewer #2 (Remarks to the Author):

The authors have done an excellent job of addressing the final concerns I raised, uncovering an artifact in the laser speckle data due to emitted fluorescence. The new data on the impact of the particles on blood flow clearly show that these tracers do not have a large impact on flow, alleviating my earlier concerns and making this approach a powerful one for in vivo studies of microcirculation.

The authors took additional data on the impact of the particles on cerebral blood flow using nonlinear and wide field microscopy. As they suggest, I recommend these data be included as an additional supplementary figure to complement the results shown in Supp. Fig. 10.

Point-by-point response to the Reviewers' comments

Reviewer #2

The authors have done an excellent job of addressing the final concerns I raised, uncovering an artifact in the laser speckle data due to emitted fluorescence. The new data on the impact of the particles on blood flow clearly show that these tracers do not have a large impact on flow, alleviating my earlier concerns and making this approach a powerful one for in vivo studies of microcirculation.

The authors took additional data on the impact of the particles on cerebral blood flow using nonlinear and wide field microscopy. As they suggest, I recommend these data be included as an additional supplementary figure to complement the results shown in Supp. Fig. 10.

Reply: We thank the Reviewer for appreciating our efforts and for recommending publication of our article. As suggested, we include the data provided in the previous review round as supplementary information (Suppl. Fig. 10 and Suppl. Fig. 11).